# Cholinesterase Deficiency Syndrome—A Pitfall in the Use of Butyrylcholinesterase as a Biomarker for Wilson’s Disease

**DOI:** 10.3390/biom12101398

**Published:** 2022-09-30

**Authors:** Max Arslan, Max Novak, Dietmar Rosenthal, Christian J. Hartmann, Philipp Albrecht, Sara Samadzadeh, Harald Hefter

**Affiliations:** 1Departments of Neurology, University of Düsseldorf, Moorenstrasse 5, 40225 Düsseldorf, Germany; 2Departments of Anesthesiology, University of Düsseldorf, Moorenstrasse 5, 40225 Düsseldorf, Germany

**Keywords:** Wilson’s disease, cholinesterase deficiency, biomarker, therapy monitoring

## Abstract

A family is described as having two recessively inherited metabolic diseases and three differently affected children. During the explantation of a drain tube grommet under general anesthesia, a prolonged resuscitation and wake-up period occurred in the key case when he was 8 years old. This led to a family screening for butyrylcholinesterase deficiency, which was confirmed not only in the key case but also in his 5-year-old sister; it was not confirmed in his 10-year-old brother. However, the key case not only had reduced serum levels of BCHE, but also elevated liver enzyme levels, which are atypical for BCHE deficiency. After the exclusion of viral and autoimmune hepatitis, Wilson’s disease (WD) was eventually diagnosed and also confirmed in his elder brother, but not in his sister. This family is presented to highlight an extremely rare WD-patient in whom a low serum level of BCHE did not occur because of WD but because of BCHE deficiency.

## 1. Introduction

Butyrylcholinesterase (BCHE), an alpha-glycoprotein (also called pseudocholinesterase or serum cholinesterase), is present in a variety of cells, especially in the hepatocytes [1]. Acute or chronic liver damage decreases the serum levels of BCHE. Therefore, BCHE has become a sensitive biomarker for liver cirrhosis [2], for patients with acute and chronic heart failure [3,4], and for those who work with pesticides [5]. The serum levels of BCHE are inversely related to COVID-19 pneumonia severity and mortality [6].

There are patients with an inborn deficiency of BCHE and low serum levels of BCHE. These patients usually live a normal life, but in cases of local or general anesthesia, the duration of effect with respect to wake-up times may become considerably prolonged [7,8]. A normal dose of muscle relaxant results in a prolonged inability to breathe [9]. Additionally, the effect of cocaine or heroin becomes life-threatening and persistent [10]. In these patients, the serum levels of BCHE are 10- to 1000-fold decreased compared to normal values [11]. A patient who knows his deficiency status can avoid prolonged apnea if the physician administers a 10-fold lower dose of muscle relaxant.

WD is a rare disorder of copper metabolism affecting the liver and brain [12]. The copper mediated liver dysfunction does lower the hepatic protein synthetic capacity and leads to lower levels not only of albumin but also of BCHE. When a population of 20 de novo WD patients was compared with 31 patients with suspected, but not confirmed, WD, the correct classification of these 51 patients was achieved in 96% of cases based on BCHE serum levels only. Among 27 parameters of blood and urine, BCHE turned out to be the parameter with the largest area under the curve in a receiver operating characteristic (ROC) curve analysis (testing sensitivity and specificity simultaneously) to predict the correct diagnosis of WD [13]. Therefore, the use of BCHE as a biomarker for Wilson’s disease (WD) has been proposed [13].

However, an obstacle to using BCHE as a biomarker of WD is patients with the rare combination of BCHE deficiency and WD. Here, for the first time, a family is described in which both BCHE deficiency and WD are inherited.

## 2. Case Reports

In the out-patient department of the university hospital in Düsseldorf (Germany), the course of treatment of more than 100 WD patients is routinely monitored. After a clinical neurological examination, patients go to the laboratory of the clinic. Blood samples are taken there immediately before the analysis. About 27 biochemical parameters are determined for therapy monitoring. In the present manuscript, only the values of AST, ALT, the serum levels of copper, 24 h urinary copper excretion and the serum levels of BCHE are presented (Figure 1A–E) for three patients.

**(1)** 
**Key Case 1 (male, age at recruitment: 18 years, BCHE-deficiency and WD)**


During early childhood, this boy suffered from chronic otitis media, and a drain tube grommet was implanted. When the child was 8 years old, the explantation of the grommet tube was performed under general anesthesia. Because of a significantly prolonged resuscitation and wake-up period, butyrylcholinesterase deficiency was immediately suspected and confirmed by testing dibucaine inhibition.

In a detailed analysis of his blood and urine, increased levels of liver enzymes were detected in addition to low levels of BCHE. This atypical finding for BCHE-deficiency led to further analysis of possible reasons for liver damage. Viral and autoimmune hepatitis could be excluded. Finally, Wilson’s disease was confirmed because of an elevated 24 h urinary copper excretion and abnormally low serum levels of copper and ceruloplasmin. Treatment with D-penicillamine (DPA), which forms copper chelating complexes and thus leads to urinary copper excretion, was initiated with 300 mg and increased up to 600 mg later on. Under this medication, the patient remained asymptomatic during the following years, and the liver enzymes became normal (full circles in Figure 1A,B). Yet, the serum copper and BCHE remained outside of the normal range (Figure 1C,E) despite a good compliance, as can be seen from the low levels of the 24 h urinary copper excretion.

**(2)** 
**Case 2 (male, age at recruitment: 20 years, WD)**


After BCHE-deficiency and WD had been diagnosed in the key case, a family screening was performed. In the second male of the family, a 10-year-old child, BCHE-deficiency was excluded due to a normal dibucaine number. Wilson’s disease was diagnosed because of increased liver enzyme levels and an elevated 24 h urinary copper excretion but decreased serum levels of ceruloplasmin. Under medication of 600 mg DPA, he also remained asymptomatic, although he had up to 20 kg more body weight compared to his brother. AST and ALT did not normalize completely (open circles in Figure 1A,B).

**(3)** 
**Case 3 (female, age at recruitment: 14 years, BCHE-deficiency)**


When BCHE-deficiency was diagnosed in case 1, his sister was 5-year-old. Screening for BCHE-deficiency revealed a low serum level of BCHE and an abnormal dibucaine number, confirming BCHE-deficiency in her as well. The girl was clinically asymptomatic but had slightly reduced serum levels of ceruloplasmin and copper (open squares in Figure 1C), which is typical for a heterozygous WD gene carrier. Her serum level of BCHE was as low as that of the key case and about 5–6-fold lower than normal values (open squares in Figure 1E).

When case 2 reached the age of 18 years, both brothers were referred to the outpatient department for WD patients at the University of Düsseldorf (Germany). Both patients were still neurologically asymptomatic. Doses of D-penicillamine were increased from 600 to 1200 mg, which caused a transient increase in 24 h urinary copper excretion in both patients (open and closed circles in Figure 1D). This was more pronounced in the elder brother. Switching to 1500 mg of Trientine^®^ induced the second peak of copper excretion in case 2 later on (open circles in Figure 1D).

## 3. Discussion

A variety of variants of BCHE have been described previously, but all resulted from mutations of one single gene [9,14]. This clarification of the origin of different BCHE variants was one of the breakthroughs of pharmacogenetics. Until 2012, about 70 natural mutations of the human BCHE gene had been documented [15]. In the European population, 1 out of 25 people is a carrier of the atypical D70G mutation on chromosome 3q26 and 1 out of 2500 have two copies of the D70G mutation [9,16]. In subpopulations with more frequent intermarriage amongst close blood relatives, the prevalence of BCHE-deficiency may be higher [17].

In daily life, BCHE-deficiency does not have significant implications. In most cases, BCHE-deficiency is diagnosed when postoperative complications occur. However, it would be much better to diagnose BCHE-deficiency early and inform patients about the complications of BCHE-deficiency, among which are prolonged awakening after electroconvulsive therapy (ECT; [18]) and cardiac arrest after cocaine abuse [10], which usually do not take place in a well-equipped surgery center.

Wilson’s disease (WD) is also thought to be a monogenetically caused metabolic disorder, with more than 1000 mutations of the ATP7B gene on chromosome 13q14.3 [19,20,21]. However, case reports of homozygotic twins with completely different clinical presentations [22,23,24] indicate an additional epigenetic influence on clinical manifestation and outcome in WD [23,24]. The prevalence of WD is estimated to be around 1:30,000 [19].

Since BCHE-deficiency results from mutations of chromosome 3q26 [14], and WD from mutations of a gene on chromosome 13q14.3 [19,20], the independent inheritance of BCHE-deficiency and WD can be assumed, resulting in an estimation of frequency of the combination of BCHE-deficiency and WD of about 1:1,000,000. This implies that about 80 patients with WD and BCHE-deficiency can be expected in Germany. However, to our knowledge, no family with this disease combination has been reported so far worldwide. Our explanation is that apparently there is no need for an additional analysis of BCHE deficiency in WD patients with low serum levels of BCHE, since liver dysfunction as an essential aspect of WD offers an obvious reason for this.

In our experience, the combination of WD and BCHE-deficiency does not lead to specific complications in daily life. In case 1, BCHE remained low despite effective copper chelating therapy (see Figure 1) during a follow-up period of about 5 years. Therefore, BCHE could not be used to monitor the recovery from the copper storage of the liver or to control adherence to therapy in this patient.

In more than 50% of patients with de novo WD, the serum level of BCHE may be as low as the levels in patients with BCHE-deficiency [13,23]. After the initiation of WD-specific therapy, BCHE significantly increases. This recovery of BCHE should be monitored carefully so as not to overlook the D70G mutant of BCHE. All WD patients with persistent, abnormally low BCHE levels, despite sufficiently high copper chelating therapy, should be tested for the presence of BCHE-deficiency because of important implications in the future in case of a positive test result. We expect that, with a more frequent application of BCHE as a biomarker in WD to monitor compliance and adherence to therapy [13], more families will be detected with the combination of the D70G mutation and mutations of the Wilson gene.

## Figures and Tables

**Figure 1 biomolecules-12-01398-f001:**
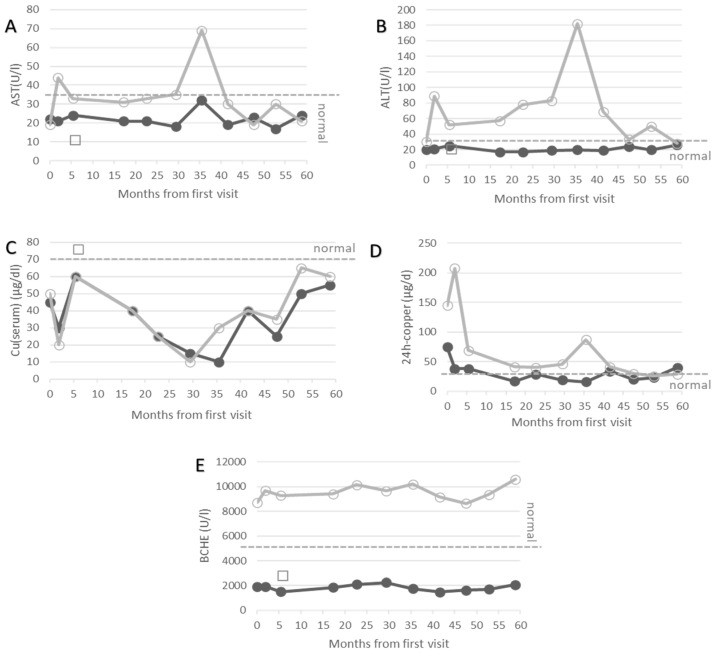
Temporal development of AST (**A**), ALT (**B**), serum levels of copper (**C**), 24 h urinary copper excretion (**D**) and butyrylcholinesterase (**E**) after the first visit in our institution. Data of the key case are presented as full circles, of the elder brother (case 2) as open circles, and of the younger sister (case 3) as open squares. The BCHE of case 1 remains low and outside the normal range at all points in time. Lower or upper limits of normal ranges of AST, ALT, Cu(serum), 24 h-copper and BCHE are indicated by grey dashed line.

## Data Availability

The data presented in this study are available on request from the corresponding author.

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
