# Peer review of "Cholinesterase Deficiency Syndrome—A Pitfall in the Use of Butyrylcholinesterase as a Biomarker for Wilson’s Disease"

_biomolecules, 2022, doi:10.3390/biom12101398_

Round 1

Reviewer 1 Report

The article  propose use of BCHE as a biomarker of Wilson's disease. The problem is that's BCHE is not specific for WD, and decrease of it is observed in different liver disorders. As we have several biomarkers, scales assessing the liver function in WD (including simply Nazer scale, MELD/PELD, KIngs College scale) as well as new imaging techniques like elastography. I don't feel that's assessment of BCHE provide any new data for monitoring of WD liver status 

Author Response

The article  propose use of BCHE as a biomarker of Wilson's disease. The problem is that's BCHE is not specific for WD, and decrease of it is observed in different liver disorders. As we have several biomarkers, scales assessing the liver function in WD (including simply Nazer scale, MELD/PELD, KIngs College scale) as well as new imaging techniques like elastography. I don't feel that's assessment of BCHE provide any new data for monitoring of WD liver status 

Sorry, the title of the manuscript was misleading. A much better title would have been:

“Cholinesterase deficiency syndrome – a pitful in the use of butyrylcholinesterase as a biomarker for Wilson´s disease”

as suggested by reviewer 2.

Reviewer 2 Report

Dear Authors,

 The idea of the research and the article presented for review is very interesting, but requires improvement a few details: 

1.       Title - I suggest changing the title as it is confusing at current form. My suggestion for consideration “Cholinesterase deficiency syndrome - a pitfall in the use of Butyrylcholinesterase as a biomarker for Wilson’s disease” 

2.       Butyrylcholinesterase (BCHE) is not a routinely used marker in laboratories to diagnose Wilson's disease. The authors cite as evidence their research, which (as I understand from the description in the bibliography) has not yet been approved for publication. Therefore, it is difficult to evaluate the correctness of the applied method of BCHE concentration assessment. I suggest that the article contain some methodological information regarding the BCHE assessment - whether the blood was tested immediately after collection or stored (under what conditions), what method was used, etc.

 3.       Case 3 was a 14-year-old or a 5-year-old girl? The current description is misleading.

 4.       Were urine copper levels tested in patient/case 3? If so, this is missing from Fig. 1D.

5.       The open squares describing case 3 should have the same size on all parts of figure 1. Please to correct it.

Sincerely yours.

Author Response

Dear Authors,

 The idea of the research and the article presented for review is very interesting, but requires improvement a few details: 

1.       Title - I suggest changing the title as it is confusing at current form. My suggestion for consideration “Cholinesterase deficiency syndrome - a pitfall in the use of Butyrylcholinesterase as a biomarker for Wilson’s disease” 

2.       Butyrylcholinesterase (BCHE) is not a routinely used marker in laboratories to diagnose Wilson's disease. The authors cite as evidence their research, which (as I understand from the description in the bibliography) has not yet been approved for publication. Therefore, it is difficult to evaluate the correctness of the applied method of BCHE concentration assessment. I suggest that the article contain some methodological information regarding the BCHE assessment - whether the blood was tested immediately after collection or stored (under what conditions), what method was used, etc.

 3.       Case 3 was a 14-year-old or a 5-year-old girl? The current description is misleading.

 4.       Were urine copper levels tested in patient/case 3? If so, this is missing from Fig. 1D.

5.       The open squares describing case 3 should have the same size on all parts of figure 1. Please to correct it.

Sincerely yours.

Reviewer 2 is absolutely right:

We have changed the title as suggested.

In our institution BCHE is routinely used for monitoring the course of treatment in WD.

After clinical examination WD-patients go to the laboratory of the clinic and blood samples are taken there immediately before the test. That is now shortly mentioned in the revised manuscript before the cases are presented.

Case 3 is a girl that was 5 years old when BCHE-deficiency was diagnosed. This uncertainty is made clear now.

Case 3 presented only once in our institution. She did not bring 24-h urine with her. Therefore the 24h-urinary copper excretion was not determined.

In the revised manuscript the open squares now have the same size in all parts of Fig. 1.

Reviewer 3 Report

This study is trying to show that plasma/serum BCHE levels are not a suitable marker for WD.

Unfortunately, the premise of the study is compromised when the report is describing a few cases, with not enough power to undermine the established test. In addition, the study contains many many scientific inaccuracies that need to be fixed. For instance, it reads that BCHE is synthesized by the liver when actually is not only detected in blood plasma (at protein level) but also is present in most cells except erythrocytes. The actual issue is that the copper-mediated liver dysfunction does lower the hepatic protein synthetic capacity. As such, not only lower circulating BCHE levels are detected but also albumin and others.

Finally, and I cannot emphasize this enough, the data mentioned in the text need to be shown. Indicating that (for instance) liver levels of BCHE were high without showing the actual values and 95%CI reference range is not acceptable. The same goes for all other data. Figures do not have error bars, it is not clear if there were technical or biological replicates, no description of any method to evaluate BCHE or any quality controls, among others.

In addition, the English grammar and use need to be improved substantially. Some sentences are very difficult to follow, and most of the verbs do not match the noun, among others. Once this is done, then the authors would need to work on the flow of the study.

Ethical concerns in regards to the patients' consent are raised as well. The authors indicate that the patients (now adults) consent to the study but actually they were children when the study started. In addition, it is mentioned that no funding was obtained; however, under the authors' contributions, a name is associated with funding acquisition.

Author Response

This study is trying to show that plasma/serum BCHE levels are not a suitable marker for WD.

Unfortunately, the premise of the study is compromised when the report is describing a few cases, with not enough power to undermine the established test. In addition, the study contains many many scientific inaccuracies that need to be fixed. For instance, it reads that BCHE is synthesized by the liver when actually is not only detected in blood plasma (at protein level) but also is present in most cells except erythrocytes. The actual issue is that the copper-mediated liver dysfunction does lower the hepatic protein synthetic capacity. As such, not only lower circulating BCHE levels are detected but also albumin and others.

Finally, and I cannot emphasize this enough, the data mentioned in the text need to be shown. Indicating that (for instance) liver levels of BCHE were high without showing the actual values and 95%CI reference range is not acceptable. The same goes for all other data. Figures do not have error bars, it is not clear if there were technical or biological replicates, no description of any method to evaluate BCHE or any quality controls, among others.

In addition, the English grammar and use need to be improved substantially. Some sentences are very difficult to follow, and most of the verbs do not match the noun, among others. Once this is done, then the authors would need to work on the flow of the study.

Ethical concerns in regards to the patients' consent are raised as well. The authors indicate that the patients (now adults) consent to the study but actually they were children when the study started. In addition, it is mentioned that no funding was obtained; however, under the authors' contributions, a name is associated with funding acquisition.

Sorry, that the manuscript had not been clear in this respect. We are convinced that BCHE is an excellent biomarker for untreated WD. But the present manuscript deals with a pitfall in the use of BCHE as a biomarker in WD. Therefore the title is changed as suggested by one of the reviewers.

We ate thankful for these corrections. Part of them have been integrated in the manuscript. We hope that reviewer 3 agrees with that.

Individual data of 3 patients are presented and no mean values. Therefore no error bars and no confidence intervals can be presented. The normal ranges defined by our University hospital laboratory are presented as light gray areas in Fig. 5.

Based on your thoughtful suggestion, aside from the correction by our native speaker, we sent the manuscript to MDPI editing system for further editing.

As long as we use pseudonymized data the local ethics committee allows publication of retrospectively collected data.

For the present study no funding was obtained. This is emphasized more explicitly.

Reviewer 4 Report

Summary:  A family with Wilson’s disease was found to carry a mutation for butyrylcholinesterase deficiency.  This rare combination of Wilson’s disease and butyrylcholinesterase mutation has not been previously reported.  The butyrylcholinesterase mutation causes complications in surgical procedures that use the muscle relaxants succinylcholine or mivacurium.  A normal dose of muscle relaxant results in prolonged inability to breathe.  Patients with the D70G mutation require assisted ventilation.  A patient who knows his butyrylcholinesterase deficiency status can avoid prolonged apnea if the physician administers a 10-fold lower dose of muscle relaxant.

1.      BChE-D is not an acceptable name for atypical BChE.  It is suggested to define the mutation D70G or Asp70Gly. 

2.     In the European population 1 out of 25 people is a carrier of the atypical D70G mutation and 1 out of 2,500 has two copies of the D70G mutation.  (Lockridge, Norgren et al. 2016)

3.     Figure 1C.  The x axis has no title.

4.     Line 109 it is suggested to replace “not oversaw” with the words “so as not to overlook the D70G mutant of BChE. 

Lockridge, O., R. B. Norgren, Jr., R. C. Johnson and T. A. Blake (2016). "Naturally Occurring Genetic Variants of Human Acetylcholinesterase and Butyrylcholinesterase and Their Potential Impact on the Risk of Toxicity from Cholinesterase Inhibitors." Chem Res Toxicol 29(9): 1381-1392.

Author Response

Summary:  A family with Wilson’s disease was found to carry a mutation for butyrylcholinesterase deficiency.  This rare combination of Wilson’s disease and butyrylcholinesterase mutation has not been previously reported.  The butyrylcholinesterase mutation causes complications in surgical procedures that use the muscle relaxants succinylcholine or mivacurium.  A normal dose of muscle relaxant results in prolonged inability to breathe.  Patients with the D70G mutation require assisted ventilation.  A patient who knows his butyrylcholinesterase deficiency status can avoid prolonged apnea if the physician administers a 10-fold lower dose of muscle relaxant.

1.      BChE-D is not an acceptable name for atypical BChE.  It is suggested to define the mutation D70G or Asp70Gly. 

2.     In the European population 1 out of 25 people is a carrier of the atypical D70G mutation and 1 out of 2,500 has two copies of the D70G mutation.  (Lockridge, Norgren et al. 2016)

3.     Figure 1C.  The x axis has no title.

4.     Line 109 it is suggested to replace “not oversaw” with the words “so as not to overlook the D70G mutant of BChE. 

Lockridge, O., R. B. Norgren, Jr., R. C. Johnson and T. A. Blake (2016). "Naturally Occurring Genetic Variants of Human Acetylcholinesterase and Butyrylcholinesterase and Their Potential Impact on the Risk of Toxicity from Cholinesterase Inhibitors." Chem Res Toxicol 29(9): 1381-1392.

This is an excellent summary of the manuscript. We hope that reviewer 4 agrees that we have integrated parts of it into the revised manuscript.

We now avoid BCHE-D.

This part is integrated.

Fig. 1C is improved.

This is corrected.

This article is cited now.

Round 2

Reviewer 1 Report

The authors making correction, changing title and article,  improved the paper, so I have no more comments  

Author Response

thanks for your helpful comments 

Reviewer 3 Report

Very few modifications were introduced in the text resulting in the same study with the same issues not corrected at all. It is not possible to claim that BCHE is a wonderful biomarker for WD with 3 cases. There's simply not enough data in this study to claim that BCDHE is a biomarker for WD. To do that, you'd need a larger n, comparison to other more traditional markers, ROC analysis etc. Figure 5 (cited in the response to the reviewer) is missing (there's only 1 figure).

Author Response

Very few modifications were introduced in the text resulting in the same study with the same issues not corrected at all.

It is not possible to claim that BCHE is a wonderful biomarker for WD with 3 cases. There's simply not enough data in this study to claim that BCDHE is a biomarker for WD. To do that, you'd need a larger n, comparison to other more traditional markers, ROC analysis etc.

Figure 5 (cited in the response to the reviewer) is missing (there's only 1 figure).

We have tried to modify the manuscript according to reviewers´ comments.

Nowhere in the manuscript we claim that BCHE is a wonderful biomarker for WD (with 3 cases).

We refer to a study with much more patients, which compares patients with definite WD and patients with suspected WD, compares BCHE with traditional markers and contains a ROC analysis.

That Fig. 5 was mentioned in the response to reviewers is an obvious mistake.

It has to be Fig. 1.
